# Off-Policy Evaluation via Off-Policy Classification

**Alex Irpan**[1], **Kanishka Rao**[1], **Konstantinos Bousmalis**[2],
**Chris Harris**[1], **Julian Ibarz**[1], **Sergey Levine**[1,3]

[1]Google Brain, Mountain View, USA
[2]DeepMind, London, UK
[3]University of California Berkeley, Berkeley, USA

`{alexirpan,kanishkarao,konstantinos,ckharris,julianibarz,slevine}@google.com`

## Abstract

In this work, we consider the problem of model selection for deep reinforcement learning (RL) in real-world environments. Typically, the performance of deep RL algorithms is evaluated via on-policy interactions with the target environment. However, comparing models in a real-world environment for the purposes of early stopping or hyperparameter tuning is costly and often practically infeasible. This leads us to examine off-policy policy evaluation (OPE) in such settings. We focus on OPE for value-based methods, which are of particular interest in deep RL, with applications like robotics, where off-policy algorithms based on Q-function estimation can often attain better sample complexity than direct policy optimization. Existing OPE metrics either rely on a model of the environment, or the use of importance sampling (IS) to correct for the data being off-policy. However, for high-dimensional observations, such as images, models of the environment can be difficult to fit and value-based methods can make IS hard to use or even ill-conditioned, especially when dealing with continuous action spaces. In this paper, we focus on the specific case of MDPs with continuous action spaces and sparse binary rewards, which is representative of many important real-world applications. We propose an alternative metric that relies on neither models nor IS, by framing OPE as a positive-unlabeled (PU) classification problem with the Q-function as the decision function. We experimentally show that this metric outperforms baselines on a number of tasks. Most importantly, it can reliably predict the relative performance of different policies in a number of generalization scenarios, including the transfer to the real-world of policies trained in simulation for an image-based robotic manipulation task.

## 1   Introduction

Supervised learning has seen significant advances in recent years, in part due to the use of large, standardized datasets [6]. When researchers can evaluate real performance of their methods on the same data via a standardized offline metric, the progress of the field can be rapid. Unfortunately, such metrics have been lacking in reinforcement learning (RL). Model selection and performance evaluation in RL are typically done by estimating the average on-policy return of a method in the target environment. Although this is possible in most simulated environments [3, 4, 37], real-world environments, like in robotics, make this difficult and expensive [36]. Off-policy evaluation (OPE) has the potential to change that: a robust off-policy metric could be used together with realistic and complex data to evaluate the expected performance of off-policy RL methods, which would enable

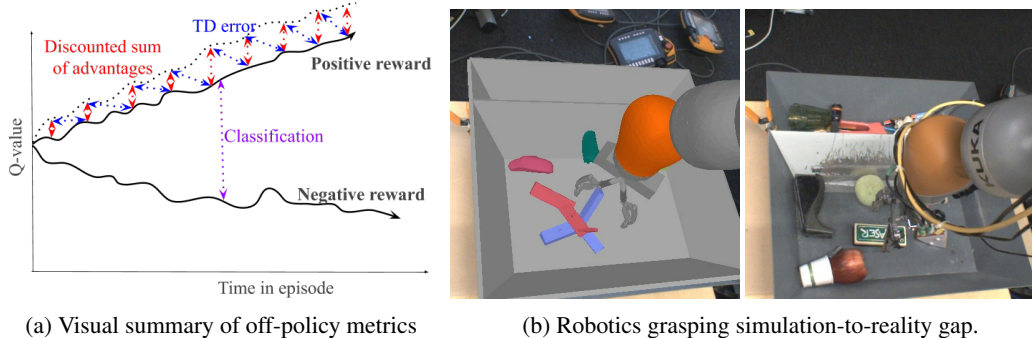

(a) Visual summary of off-policy metrics                    (b) Robotics grasping simulation-to-reality gap.

Figure 1: **(a) Visual illustration of our method:** We propose using classification-based approaches to do off-policy evaluation. Solid curves represent $Q(\mathbf{s}, \mathbf{a})$ over a positive and negative trajectory, with the dashed curve representing $\max_{\mathbf{a}'} Q(\mathbf{s}, \mathbf{a}')$ along states the positive trajectory visits (the corresponding negative curve is omitted for simplicity). Baseline approaches (blue, red) measure Q-function fit between $Q(\mathbf{s}, \mathbf{a})$ to $\max_{\mathbf{a}'} Q(\mathbf{s}, \mathbf{a}')$. Our approach (purple) directly measures separation of $Q(\mathbf{s}, \mathbf{a})$ between positive and negative trajectories. **(b) The visual "reality gap" of our most challenging task:** off-policy evaluation of the generalization of image-based robotic agents trained solely in simulation (left) using historical data from the target real-world environment (right).

rapid progress on important real-world RL problems. Furthermore, it would greatly simplify transfer learning in RL, where OPE would enable model selection and algorithm design in simple domains (e.g., simulation) while evaluating the performance of these models and algorithms on complex domains (e.g., using previously collected real-world data).

Previous approaches to off-policy evaluation [7, 13, 28, 35] generally use importance sampling (IS) or learned dynamics models. However, this makes them difficult to use with many modern deep RL algorithms. First, OPE is most useful in the off-policy RL setting, where we expect to use real-world data as the "validation set", but many of the most commonly used off-policy RL methods are based on value function estimation, produce deterministic policies [20, 38], and do not require any knowledge of the policy that generated the real-world training data. This makes them difficult to use with IS. Furthermore, many of these methods might be used with high-dimensional observations, such as images. Although there has been considerable progress in predicting future images [2, 19], learning sufficiently accurate models in image space for effective evaluation is still an open research problem. We therefore aim to develop an OPE method that requires neither IS nor models.

We observe that for model selection, it is sufficient to predict some statistic *correlated* with policy return, rather than directly predict policy return. We address the specific case of binary-reward MDPs: tasks where the agent receives a non-zero reward only once during an episode, at the final timestep (Sect. 2). These can be interpreted as tasks where the agent can either "succeed" or "fail" in each trial, and although they form a subset of all possible MDPs, this subset is quite representative of many real-world tasks, and is actively used e.g. in robotic manipulation [15, 31]. The novel contribution of our method (Sect. 3) is to frame OPE as a positive-unlabeled (PU) classification [16] problem, which provides for a way to derive OPE metrics that are both *(a)* fundamentally different from prior methods based on IS and model learning, and *(b)* perform well in practice on both simulated and real-world tasks. Additionally, we identify and present (Sect. 4) a list of generalization scenarios in RL that we would want our metrics to be robust against. We experimentally show (Sect. 6) that our suggested OPE metrics outperform a variety of baseline methods across all of the evaluation scenarios, including a simulation-to-reality transfer scenario for a vision-based robotic grasping task (see Fig. 1b).

## 2   Preliminaries

We focus on finite–horizon Markov decision processes (MDP). We define an MDP as $(\mathcal{S}, \mathcal{A}, \mathcal{P}, \mathcal{S}_0, r, \gamma)$. $\mathcal{S}$ is the state–space, $\mathcal{A}$ the action–space, and both can be continuous. $\mathcal{P}$ defines transitions to next states given the current state and action, $\mathcal{S}_0$ defines initial state distribution, $r$ is the reward function, and $\gamma \in [0, 1]$ is the discount factor. Episodes are of finite length $T$: at a given

time-step $t$ the agent is at state $\mathbf{s}_t \in \mathcal{S}$, samples an action $\mathbf{a}_t \in \mathcal{A}$ from a policy $\pi$, receives a reward $r_t = r(\mathbf{s}_t, \mathbf{a}_t)$, and observes the next state $\mathbf{s}_{t+1}$ as determined by $\mathcal{P}$.

The goal in RL is to learn a policy $\pi(\mathbf{a}_t|\mathbf{s}_t)$ that maximizes the expected episode return $R(\pi) = \mathbb{E}_\pi[\sum_{t=0}^{T} \gamma^t r(\mathbf{s}_t, \mathbf{a}_t)]$. A value of a policy for a given state $\mathbf{s}_t$ is defined as $V^\pi(\mathbf{s}_t) = \mathbb{E}_\pi[\sum_{t'=t}^{T} \gamma^{t'-t} r(\mathbf{s}_{t'}, \mathbf{a}_{t'}^\pi)]$ where $\mathbf{a}_t^\pi$ is the action $\pi$ takes at state $\mathbf{s}_t$ and $\mathbb{E}_\pi$ implies an expectation over trajectories $\tau = (\mathbf{s}_1, \mathbf{a}_1, \ldots, \mathbf{s}_T, \mathbf{a}_T)$ sampled from $\pi$. Given a policy $\pi$, the expected value of its action $a_t$ at a state $s_t$ is called the Q-value and is defined as $Q^\pi(\mathbf{s}_t, \mathbf{a}_t) = \mathbb{E}_\pi[r(\mathbf{s}_t, \mathbf{a}_t) + V^\pi(\mathbf{s}_{t+1})]$.

We assume the MDP is a binary reward MDP, which satisfies the following properties: $\gamma = 1$, the reward is $r_t = 0$ at all intermediate steps, and the final reward $r_T$ is in $\{0, 1\}$, indicating whether the final state is a failure or a success. We learn Q-functions $Q(\mathbf{s}, \mathbf{a})$ and aim to evaluate policies $\pi(\mathbf{s}) = \arg\max_{\mathbf{a}} Q(\mathbf{s}, \mathbf{a})$.

## 2.1 Positive-unlabeled learning

Positive-unlabeled (PU) learning is a set of techniques for learning binary classification from partially labeled data, where we have many unlabeled points and some positively labeled points [16]. We will make use of these ideas in developing our OPE metric. Positive-unlabeled data is sufficient to learn a binary classifier if the positive class prior $p(y = 1)$ is known.

Let $(X, Y)$ be a labeled binary classification problem, where $Y = \{0, 1\}$. Let $g : X \to \mathbb{R}$ be some decision function, and let $\ell : \mathbb{R} \times \{0, 1\} \to \mathbb{R}$ be our loss function. Suppose we want to evaluate loss $\ell(g(x), y)$ over negative examples $(x, y = 0)$, but we only have unlabeled points $x$ and positively labeled points $(x, y = 1)$. The key insight of PU learning is that the loss over negatives can be indirectly estimated from $p(y = 1)$. For any $x \in X$,

$$p(x) = p(x|y = 1)p(y = 1) + p(x|y = 0)p(y = 0) \tag{1}$$

It follows that for any $f(x)$, $\mathbb{E}_{X,Y}[f(x)] = p(y = 1)\mathbb{E}_{X|Y=1}[f(x)] + p(y = 0)\mathbb{E}_{X|Y=0}[f(x)]$, since by definition $\mathbb{E}_X[f(x)] = \int_x p(x)f(x)dx$. Letting $f(x) = \ell(g(x), 0)$ and rearranging gives

$$p(y = 0)\mathbb{E}_{X|Y=0}[\ell(g(x), 0)] = \mathbb{E}_{X,Y}[\ell(g(x), 0)] - p(y = 1)\mathbb{E}_{X|Y=1}[\ell(g(x), 0)] \tag{2}$$

In Sect. 3, we reduce off-policy evaluation of a policy $\pi$ to a positive-unlabeled classification problem. We provide reasoning for how to estimate $p(y = 1)$, apply PU learning to estimate classification error with Eqn. 2, then use the error to estimate a lower bound on return $R(\pi)$ with Theorem 1.

## 3 Off-policy evaluation via state-action pair classification

A Q-function $Q(\mathbf{s}, \mathbf{a})$ predicts the expected return of each action $\mathbf{a}$ given state $\mathbf{s}$. The policy $\pi(\mathbf{s}) = \arg\max_{\mathbf{a}} Q(\mathbf{s}, \mathbf{a})$ can be viewed as a classifier that predicts the best action. We propose an off-policy evaluation method connecting off-policy evaluation to estimating validation error for a positive-unlabeled (PU) classification problem [16]. Our metric can be used with Q-function estimation methods without requiring importance sampling, and can be readily applied in a scalable way to image-based deep RL tasks.

We present an analysis for binary reward MDPs, defined in Sec. 2. In binary reward MDPs, each $(\mathbf{s}_t, \mathbf{a}_t)$ is either potentially effective, or guaranteed to lead to failure.

**Definition 1.** In a binary reward MDP, $(\mathbf{s}_t, \mathbf{a}_t)$ is **feasible** if an optimal policy $\pi^*$ has non-zero probability of achieving success, i.e an episode return of 1, after taking $\mathbf{a}_t$ in $\mathbf{s}_t$. A state-action pair $(\mathbf{s}_t, \mathbf{a}_t)$ is **catastrophic** if even an optimal $\pi^*$ has zero probability of succeeding if $\mathbf{a}_t$ is taken. A state $\mathbf{s}_t$ is feasible if there exists a feasible $(\mathbf{s}_t, \mathbf{a}_t)$, and a state $\mathbf{s}_t$ is catastrophic if for all actions $\mathbf{a}_t$, $(\mathbf{s}_t, \mathbf{a}_t)$ is catastrophic.

Under this definition, the return of a trajectory $\tau$ is 1 only if all $(\mathbf{s}_t, \mathbf{a}_t)$ in $\tau$ are feasible (see Appendix A.1). This condition is necessary, but not sufficient, because success can depend on the stochastic dynamics. Since Definition 1 is defined by an optimal $\pi^*$, we can view feasible and catastrophic as binary labels that are independent of the policy $\pi$ we are evaluating. Viewing $\pi$ as a classifier, we relate the classification error of $\pi$ to a lower bound for return $R(\pi)$.

**Theorem 1.** *Given a binary reward MDP and a policy $\pi$, let $c(\mathbf{s}_t, \mathbf{a}_t)$ be the probability that stochastic dynamics bring a feasible $(\mathbf{s}_t, \mathbf{a}_t)$ to a catastrophic $\mathbf{s}_{t+1}$, with $c = \max_{\mathbf{s},\mathbf{a}} c(\mathbf{s}, \mathbf{a})$. Let $\rho_{t,\pi}^+$ denote the state distribution at time $t$, given that $\pi$ was followed, all its previous actions $\mathbf{a}_1, \cdots, \mathbf{a}_{t-1}$ were feasible, and $\mathbf{s}_t$ is feasible. Let $\mathcal{A}_-(\mathbf{s})$ denote the set of catastrophic actions at state $\mathbf{s}$, and let $\epsilon_t = \mathbb{E}_{\rho_{t,\pi}^+} \left[ \sum_{\mathbf{a} \in \mathcal{A}_-(\mathbf{s}_t)} \pi(\mathbf{a}|\mathbf{s}_t) \right]$ be the per-step expectation of $\pi$ making its first mistake at time $t$, with $\epsilon = \frac{1}{T} \sum_{i=1}^{T} \epsilon_t$ being average error over all $(\mathbf{s}_t, \mathbf{a}_t)$. Then $R(\pi) \geq 1 - T(\epsilon + c)$.*

See Appendix A.2 for the proof. For the deterministic case ($c = 0$), we can take inspiration from imitation learning behavioral cloning bounds in Ross & Bagnell [32] to prove the same result. This alternate proof is in Appendix A.3.

A smaller error $\epsilon$ gives a higher lower bound on return, which implies a better $\pi$. This leaves estimating $\epsilon$. The primary challenge with this approach is that we do not have negative labels – that is, for trials that receive a return of 0 in the validation set, we do not know which $(\mathbf{s}, \mathbf{a})$ were in fact catastrophic, and which were recoverable. We discuss how we address this problem next.

### 3.1 Missing negative labels

Recall that $(\mathbf{s}_t, \mathbf{a}_t)$ is feasible if $\pi^*$ has a chance of succeeding after action $\mathbf{a}_t$. Since $\pi^*$ is at least as good as $\pi_b$, whenever $\pi_b$ succeeds, all tuples $(\mathbf{s}_t, \mathbf{a}_t)$ in the trajectory $\tau$ must be feasible. However, the converse is not true, since failure could come from poor luck or suboptimal actions. Our key insight is that this is an instance of the positive-unlabeled (PU) learning problem from Sect. 2.1, where $\pi_b$ positively labels some $(\mathbf{s}, \mathbf{a})$ and the remaining are unlabeled. This lets us use ideas from PU learning to estimate error.

In the RL setting, inputs $(\mathbf{s}, \mathbf{a})$ are from $X = \mathcal{S} \times \mathcal{A}$, labels $\{0, 1\}$ correspond to $\{catastrophic, feasible\}$ labels, and a natural choice for the decision function $g$ is $g(\mathbf{s}, \mathbf{a}) = Q(\mathbf{s}, \mathbf{a})$, since $Q(\mathbf{s}, \mathbf{a})$ should be high for feasible $(\mathbf{s}, \mathbf{a})$ and low for catastrophic $(\mathbf{s}, \mathbf{a})$. We aim to estimate $\epsilon$, the probability that $\pi$ takes a catastrophic action – i.e., that $(\mathbf{s}, \pi(\mathbf{s}))$ is a false positive. Note that if $(\mathbf{s}, \pi(\mathbf{s}))$ is predicted to be catastrophic, but is actually feasible, this false-negative does not impact future reward – since the action is feasible, there is still some chance of success. We want just the false-positive risk, $\epsilon = p(y = 0)\mathbb{E}_{X|Y=0}\left[\ell(g(x), 0)\right]$. This is the same as Eqn. 2, and using $g(\mathbf{s}, \mathbf{a}) = Q(\mathbf{s}, \mathbf{a})$ gives

$$\epsilon = \mathbb{E}_{(\mathbf{s},\mathbf{a})}\left[\ell(Q(\mathbf{s}, \mathbf{a}), 0)\right] - p(y=1)\mathbb{E}_{(\mathbf{s},\mathbf{a}), y=1}\left[\ell(Q(\mathbf{s}, \mathbf{a}), 0)\right]. \qquad (3)$$

Eqn. 3 is the core of all our proposed metrics. While it might at first seem that the class prior $p(y = 1)$ should be task-dependent, recall that the error $\epsilon_t$ is the expectation over the state distribution $\rho_{t,\pi}^+$, where the actions $\mathbf{a}_1, \cdots, \mathbf{a}_{t-1}$ were all feasible. This is equivalent to following an optimal "expert" policy $\pi^*$, and although we are estimating $\epsilon_t$ from data generated by behavior policy $\pi_b$, we should match the positive class prior $p(y = 1)$ we would observe from expert $\pi^*$. Expert $\pi^*$ will always pick feasible actions. Therefore, although the validation dataset will likely have both successes and failures, a prior of $p(y = 1) = 1$ is the ideal prior, and this holds independently of the environment. We illustrate this further with a didactic example in Sect. 6.1.

Theorem 1 relies on estimating $\epsilon$ over the distribution $\rho_{t,\pi}^+$, but our dataset $\mathcal{D}$ is generated by an unknown behavior policy $\pi_b$. A natural approach here would be importance sampling (IS) [7], but: *(a)* we assume no knowledge of $\pi_b$, and *(b)* IS is not well-defined for deterministic policies $\pi(\mathbf{s}) = \arg\max_{\mathbf{a}} Q(\mathbf{s}, \mathbf{a})$. Another approach is to subsample $\mathcal{D}$ to transitions $(\mathbf{s}, \mathbf{a})$ where $\mathbf{a} = \pi(\mathbf{s})$ [21]. This ensures an on-policy evaluation, but can encounter finite sample issues if $\pi_b$ does not sample $\pi(\mathbf{s})$ frequently enough. Therefore, we assume classification error over $\mathcal{D}$ is a good enough proxy that correlates well with classification error over $\rho_{t,\pi}^+$. This is admittedly a strong assumption, but empirical results in Sect. 6 show surprising robustness to distributional mismatch. This assumption is reasonable if $\mathcal{D}$ is broad (e.g., generated by a sufficiently random policy), but may produce pessimistic estimates when potential feasible actions in $\mathcal{D}$ are unlabeled.

### 3.2 Off-policy classification for OPE

Based off of the derivation from Sect. 3.1, our proposed off-policy classification (OPC) score is defined by the negative loss when $\ell$ in Eqn. 3 is the 0-1 loss. Let $b$ be a threshold, with

$\ell(Q(\mathbf{s}, \mathbf{a}), Y) = \frac{1}{2} + \left(\frac{1}{2} - Y\right) \text{sign}(Q(\mathbf{s}, \mathbf{a}) - b)$. This gives

$$\text{OPC}(Q) = p(y=1)\mathbb{E}_{(\mathbf{s},\mathbf{a}),y=1}\left[1_{Q(\mathbf{s},\mathbf{a})>b}\right] - \mathbb{E}_{(\mathbf{s},\mathbf{a})}\left[1_{Q(\mathbf{s},\mathbf{a})>b}\right]. \tag{4}$$

To be fair to each $Q(\mathbf{s}, \mathbf{a})$, threshold $b$ is set separately for each Q-function to maximize OPC($Q$). Given $N$ transitions and $Q(\mathbf{s}, \mathbf{a})$ for all $(\mathbf{s}, \mathbf{a}) \in \mathcal{D}$, this can be done in $O(N \log N)$ time per Q-function (see Appendix B). This avoids favoring Q-functions that systematically overestimate or underestimate the true value.

Alternatively, $\ell$ can be a soft loss function. We experimented with $\ell(Q(\mathbf{s}, \mathbf{a}), Y) = (1 - 2Y)Q(\mathbf{s}, \mathbf{a})$, which is minimized when $Q(\mathbf{s}, \mathbf{a})$ is large for $Y = 1$ and small for $Y = 0$. The negative of this loss is called the SoftOPC.

$$\text{SoftOPC}(Q) = p(y=1)\mathbb{E}_{(\mathbf{s},\mathbf{a}),y=1}\left[Q(\mathbf{s},\mathbf{a})\right] - \mathbb{E}_{(\mathbf{s},\mathbf{a})}\left[Q(\mathbf{s},\mathbf{a})\right]. \tag{5}$$

If episodes have different lengths, to avoid focusing on long episodes, transitions $(\mathbf{s}, \mathbf{a})$ from an episode of length $T$ are weighted by $\frac{1}{T}$ when estimating SoftOPC. Pseudocode is in Appendix B.

Although our derivation is for binary reward MDPs, both the OPC and SoftOPC are purely evaluation time metrics, and can be applied to Q-functions trained with dense rewards or reward shaping, as long as the final evaluation uses a sparse binary reward.

### 3.3 Evaluating OPE metrics

The standard evaluation method for OPE is to report MSE to the true episode return [21, 35]. However, our metrics do not estimate episode return directly. The OPC($Q$)'s estimate of $\epsilon$ will differ from the true value, since it is estimated over our dataset $\mathcal{D}$ instead of over the distribution $\rho^+_{t,\pi}$. Meanwhile, SoftOPC($Q$) does not estimate $\epsilon$ directly due to using a soft loss function. Despite this, the OPC and SoftOPC are still useful OPE metrics if they *correlate* well with $\epsilon$ or episode return $R(\pi)$.

We propose an alternative evaluation method. Instead of reporting MSE, we train a large suite of Q-functions $Q(\mathbf{s}, \mathbf{a})$ with different learning algorithms, evaluating true return of the equivalent argmax policy for each $Q(\mathbf{s}, \mathbf{a})$, then compare correlation of the metric to true return. We report two correlations, the coefficient of determination $R^2$ of line of best fit, and the Spearman rank correlation $\xi$ [33].[1] $R^2$ measures confidence in how well our linear best fit will predict returns of new models, whereas $\xi$ measures confidence that the metric ranks different policies correctly, without assuming a linear best fit.

## 4 Applications of OPE for transfer and generalization

Off-policy evaluation (OPE) has many applications. One is to use OPE as an early stopping or model selection criteria when training from off-policy data. Another is applying OPE to validation data collected in another domain to measure generalization to new settings. Several papers [5, 27, 30, 39, 40] have examined overfitting and memorization in deep RL, proposing explicit train-test environment splits as benchmarks for RL generalization. Often, these test environments are defined in simulation, where it is easy to evaluate the policy in the test environment. This is no longer sufficient for real-world settings, where test environment evaluation can be expensive. In real-world problems, off-policy evaluation is an inescapable part of measuring generalization performance in an efficient, tractable way. To demonstrate this, we identify a few common generalization failure scenarios faced in reinforcement learning, applying OPE to each one. When there is *insufficient off-policy training data* and new data is not collected online, models may memorize state-action pairs in the training data. RL algorithms collect new on-policy data with high frequency. If training data is generated in a systematically biased way, we have *mismatched off-policy training data*. The model fails to generalize because systemic biases cause the model to miss parts of the target distribution. Finally, models trained in simulation usually do not generalize to the real-world, due to the *training and test domain gap*: the differences in the input space (see Fig. 1b and Fig. 2) and the dynamics. All of these scenarios are, in principle, identifiable by off-policy evaluation, as long as validation is done against data sampled from the final testing environment. We evaluate our proposed and baseline metrics across all these scenarios.

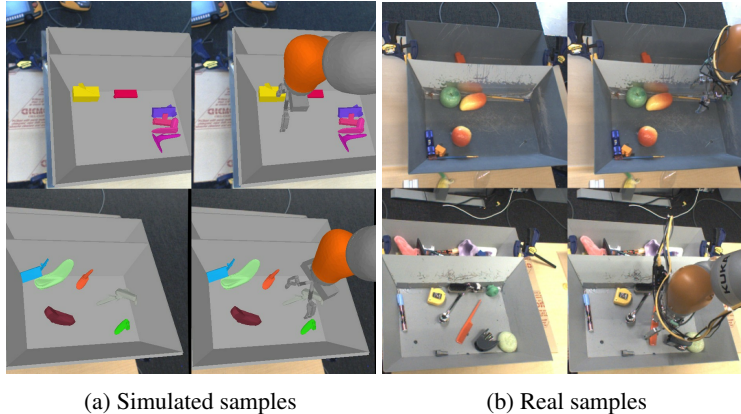

| (a) Simulated samples | (b) Real samples |

Figure 2: **An example of a training and test domain gap.** We display this with a robotic grasping task. *Left:* Images used during training, from (a) simulated grasping over procedurally generated objects; and from (b) the real-world, with a varied collection of everyday physical objects.

## 5 Related work

Off-policy policy evaluation (OPE) predicts the return of a learned policy $\pi$ from a fixed off-policy dataset $\mathcal{D}$, generated by one or more behavior policies $\pi_b$. Prior works [7, 10, 13, 21, 28, 34] do so with importance sampling (IS) [11], MDP modeling, or both. Importance sampling requires querying $\pi(\mathbf{a}|\mathbf{s})$ and $\pi_b(\mathbf{a}|\mathbf{s})$ for any $\mathbf{s} \in \mathcal{D}$, to correct for the shift in state-action distributions. In RL, the cumulative product of IS weights along $\tau$ is used to weight its contribution to $\pi$'s estimated value [28]. Several variants have been proposed, such as step-wise IS and weighted IS [23]. In MDP modeling, a model is fitted to $\mathcal{D}$, and $\pi$ is rolled out in the learned model to estimate average return [13, 24]. The performance of these approaches is worse if dynamics or reward are poorly estimated, which tends to occur for image-based tasks. Improving these models is an active research question [2, 19]. State of the art methods combine IS-based estimators and model-based estimators using doubly robust estimation and ensembles to produce improved estimators with theoretical guarantees [7, 8, 10, 13, 35].

These IS and model-based OPE approaches assume importance sampling or model learning are feasible. This assumption often breaks down in modern deep RL approaches. When $\pi_b$ is unknown, $\pi_b(\mathbf{a}|\mathbf{s})$ cannot be queried. When doing value-based RL with deterministic policies, $\pi(\mathbf{a}|\mathbf{s})$ is undefined for off-policy actions. When working with high-dimensional observations, model learning is often too difficult to learn a reliable model for evaluation.

Many recent papers [5, 27, 30, 39, 40] have defined train-test environment splits to evaluate RL generalization, but define test environments in simulation where there is no need for OPE. We demonstrate how OPE provides tools to evaluate RL generalization for real-world environments. While to our knowledge no prior work has proposed a classification-based OPE approach, several prior works have used supervised classifiers to predict transfer performance from a few runs in the test environment [17, 18]. To our knowledge, no other OPE papers have shown results for large image-based tasks where neither importance sampling nor model learning are viable options.

**Baseline metrics** Since we assume importance-sampling and model learning are infeasible, many common OPE baselines do not fit our problem setting. In their place, we use other Q-learning based metrics that also do not need importance sampling or model learning and only require a $Q(\mathbf{s}, \mathbf{a})$ estimate. The *temporal-difference error* (TD Error) is the standard Q-learning training loss, and Farahmand & Szepesvári [9] proposed a model selection algorithm based on minimizing TD error. The *discounted sum of advantages* ($\sum_t \gamma^t A^\pi$) relates the difference in values $V^{\pi_b}(\mathbf{s}) - V^\pi(\mathbf{s})$ to the sum of advantages $\sum_t \gamma^t A^\pi(\mathbf{s}, \mathbf{a})$ over data from $\pi_b$, and was proposed by Kakade & Langford [14] and Murphy [26]. Finally, the *Monte Carlo corrected error* (MCC Error) is derived by arranging the discounted sum of advantages into a squared error, and was used as a training objective by Quillen et al. [29]. The exact expression of each of these metrics is in Appendix C.

Each of these baselines represents a different way to measure how well $Q(\mathbf{s}, \mathbf{a})$ fits the true return. However, it is possible to learn a good policy $\pi$ even when $Q(\mathbf{s}, \mathbf{a})$ fits the data poorly. In Q-learning,

it is common to define an argmax policy $\pi = \arg\max_{\mathbf{a}} Q(\mathbf{s}, \mathbf{a})$. The argmax policy for $Q^*(\mathbf{s}, \mathbf{a})$ is $\pi^*$, and $Q^*$ has zero TD error. But, applying any monotonic function to $Q^*(\mathbf{s}, \mathbf{a})$ produces a $Q'(\mathbf{s}, \mathbf{a})$, whose TD error is non-zero, but whose argmax policy is still $\pi^*$. A good OPE metric should rate $Q^*$ and $Q'$ identically. This motivates our proposed classification-based OPE metrics: since $\pi$'s behavior only depends on the relative differences in Q-value, it makes sense to directly contrast Q-values against each other, rather than compare error between the Q-values and episode return. Doing so lets us compare Q-functions whose $Q(\mathbf{s}, \mathbf{a})$ estimates are inaccurate. Fig. 1a visualizes the differences between the baseline metrics and classification metrics.

# 6 Experiments

In this section, we investigate the correlation of OPC and SoftOPC with true average return, and how they may be used for model selection with off-policy data. We compare the correlation of these metrics with the correlation of the baselines, namely the TD Error, Sum of Advantages, and the MCC Error (see Sect. 5) in a number of environments and generalization failure scenarios. For each experiment, a validation dataset $\mathcal{D}$ is collected with a behavior policy $\pi_b$, and state-action pairs $(\mathbf{s}, \mathbf{a})$ are labeled as feasible whenever they appear in a successful trajectory. In line with Sect. 3.3, several Q-functions $Q(\mathbf{s}, \mathbf{a})$ are trained for each task. For each $Q(\mathbf{s}, \mathbf{a})$, we evaluate each metric over $\mathcal{D}$ and true return of the equivalent argmax policy. We report both the coefficient of determination $R^2$ of line of best fit and the Spearman's rank correlation coefficient $\xi$ [33]. Our results are summarized in Table 1 and Table 2. Our OPC/SoftOPC metrics are implemented using $p(y = 1) = 1$, as explained in Sect. 3 and Appendix D.

## 6.1 Simple Environments

**Binary tree.** As a didactic toy example, we used a binary tree MDP with depth of episode length $T$. In this environment,[2] each node is a state $\mathbf{s}_t$ with $r_t = 0$, unless it is a leaf/terminal state with reward $r_T \in \{0, 1\}$. Actions are {'left', 'right'}, and transitions are deterministic. Exactly one leaf is a success leaf with $r_T = 1$, and the rest have $r_T = 0$. In our experiments we used a full binary tree of depth $T = 6$. The initial state distribution was uniform over all non-leaf nodes, which means that the initial state could sometimes be initialized to one where failure is inevitable. The validation dataset $\mathcal{D}$ was collected by generating 1,000 episodes from a uniformly random policy. For the policies we wanted to evaluate, we generated 1,000 random Q-functions by sampling $Q(\mathbf{s}, \mathbf{a}) \sim U[0, 1]$ for every $(\mathbf{s}, \mathbf{a})$, defining the policy as $\pi(\mathbf{s}) = \arg\max_{\mathbf{a}} Q(\mathbf{s}, \mathbf{a})$. We compared the correlation of the actual on-policy performance of the policies with the scores given by the OPC, SoftOPC and the baseline metrics using $\mathcal{D}$, as shown in Table 2. SoftOPC correlates best and OPC correlates second best.

**Pong.** As we are specifically motivated by image-based tasks with binary rewards, the Atari [3] Pong game was a good choice for a simple environment that can have these characteristics. The visual input is of low complexity, and the game can be easily converted into a binary reward task by truncating the episode after the first point is scored. We learned Q-functions using DQN [25] and DDQN [38], varying hyperparameters such as the learning rate, the discount factor $\gamma$, and the batch size, as discussed in detail in Appendix E.2. A total of 175 model checkpoints are chosen from the various models for evaluation, and true average performance is evaluated over 3,000 episodes for each model checkpoint. For the validation dataset we used 38 Q-functions that were partially-trained with DDQN and generated 30 episodes from each, for a total of 1140 episodes. Similarly with the Binary Tree environments we compare the correlations of our metrics and the baselines to the true average performance over a number of on-policy episodes. As we show in Table 2, both our metrics outperform the baselines, OPC performs better than SoftOPC in terms of $R^2$ correlation but is similar in terms of Spearman correlation $\xi$.

**Stochastic dynamics.** To evaluate performance against stochastic dynamics, we modified the dynamics of the binary tree and Pong environment. In the binary tree, the environment executes a random action instead of the policy's action with probability $\epsilon$. In Pong, the environment uses sticky actions, a standard protocol for stochastic dynamics in Atari games introduced by [22]. With small probability, the environment repeats the previous action instead of the policy's action. Everything else

is unchanged. Results in Table 1. In more stochastic environments, all metrics drop in performance since $Q(s, a)$ has less control over return, but OPC and SoftOPC consistently correlate better than the baselines.

Table 1: Results from stochastic dynamics experiments. For each metric (leftmost column), we report $R^2$ of line of best fit and Spearman rank correlation coefficient $\xi$ for each environment (top row), over stochastic versions of the binary tree and Pong tasks from Sect. 6.1. Correlation drops as stochasticity increases, but our proposed metrics (last two rows) consistently outperform baselines.

| | Stochastic Tree 1-Success Leaf | | | | | | Pong Sticky Actions | | | |
| | $\epsilon = 0.4$ | | $\epsilon = 0.6$ | | $\epsilon = 0.8$ | | Sticky 10% | | Sticky 25% | |
| | $R^2$ | $\xi$ | $R^2$ | $\xi$ | $R^2$ | $\xi$ | $R^2$ | $\xi$ | $R^2$ | $\xi$ |
|---|---|---|---|---|---|---|---|---|---|---|
| TD Err | 0.01 | -0.07 | 0.00 | -0.05 | 0.00 | -0.05 | 0.05 | -0.16 | 0.07 | -0.15 |
| $\sum \gamma^t A^\pi$ | 0.00 | 0.01 | 0.01 | -0.07 | 0.00 | -0.02 | 0.04 | -0.29 | 0.01 | -0.22 |
| MCC Err | 0.07 | -0.27 | 0.01 | -0.06 | 0.01 | -0.11 | 0.02 | -0.32 | 0.00 | -0.18 |
| OPC (Ours) | 0.13 | 0.38 | 0.01 | 0.08 | 0.03 | 0.19 | **0.48** | **0.73** | **0.33** | **0.66** |
| SoftOPC (Ours) | **0.14** | **0.39** | **0.03** | **0.18** | **0.04** | **0.20** | 0.33 | 0.67 | 0.16 | 0.58 |

## 6.2 Vision-based Robotic Grasping

Our main experimental results were on simulated and real versions of a robotic environment and a vision-based grasping task, following the setup from Kalashnikov et al. [15], the details of which we briefly summarize. The observation at each time-step is a $472 \times 472$ RGB image from a camera placed over the shoulder of a robotic arm, of the robot and a bin of objects, as shown in Fig. 1b. At the start of an episode, objects are randomly dropped in a bin in front of the robot. The goal is to grasp any of the objects in that bin. Actions include continuous Cartesian displacements of the gripper, and the rotation of the gripper around the z-axis. The action space also includes three discrete commands: "open gripper", "close gripper", and "terminate episode". Rewards are sparse, with $r(\mathbf{s}_T, \mathbf{a}_T) = 1$ if any object is grasped and $0$ otherwise. All models are trained with the fully off-policy QT-Opt algorithm as described in Kalashnikov et al. [15].

In simulation we define a training and a test environment by generating two distinct sets of 5 objects that are used for each, shown in Fig. 8. In order to capture the different possible generalization failure scenarios discussed in Sect. 4, we trained Q-functions in a fully off-policy fashion with data collected by a hand-crafted policy with a 60% grasp success rate and $\epsilon$-greedy exploration (with $\epsilon$=0.1) with two different datasets both from the training environment. The first consists of $100,000$ episodes, with which we can show we have *insufficient off-policy training data* to perform well even in the training environment. The second consists of $900,000$ episodes, with which we can show we have sufficient data to perform well in the training environment, but due to *mismatched off-policy training data* we can show that the policies do not generalize to the test environment (see Fig. 8 for objects and Appendix E.3 for the analysis). We saved policies at different stages of training which resulted in 452 policies for the former case and 391 for the latter. We evaluated the true return of these policies on 700 episodes on each environment and calculated the correlation with the scores assigned by the OPE metrics based on held-out validation sets of $50,000$ episodes from the training environment and $10,000$ episodes from the test one, which we show in Table 2.

Table 2: Summarized results of Experiments section. For each metric (leftmost column), we report $R^2$ of line of best fit and Spearman rank correlation coefficient $\xi$ for each environment (top row). These are: the binary tree and Pong tasks from Sect. 6.1, simulated grasping with train or test objects, and real-world grasping from Sect. 6.2. Baseline metrics are discussed in Sect. 5, and our metrics (OPC, SoftOPC) are discussed in Sect. 3. Occasionally, some baselines correlate well, but our proposed metrics (last two rows) are consistently among the best metrics for each environment.

| | Tree (1 Succ) | | Pong | | Sim Train | | Sim Test | | Real-World | |
| | $R^2$ | $\xi$ | $R^2$ | $\xi$ | $R^2$ | $\xi$ | $R^2$ | $\xi$ | $R^2$ | $\xi$ |
|---|---|---|---|---|---|---|---|---|---|---|
| TD Err | 0.02 | -0.15 | 0.05 | -0.18 | 0.02 | -0.37 | 0.10 | -0.51 | 0.17 | 0.48 |
| $\sum \gamma^t A^\pi$ | 0.00 | 0.00 | 0.09 | -0.32 | **0.74** | 0.81 | **0.74** | **0.78** | 0.12 | 0.50 |
| MCC Err | 0.06 | -0.26 | 0.04 | -0.36 | 0.00 | 0.33 | 0.06 | -0.44 | 0.01 | -0.15 |
| OPC (Ours) | **0.21** | 0.50 | **0.50** | 0.72 | 0.49 | **0.86** | 0.35 | 0.66 | 0.81 | 0.87 |
| SoftOPC (Ours) | 0.19 | **0.51** | 0.36 | **0.75** | 0.55 | 0.76 | 0.48 | 0.77 | **0.91** | **0.94** |

The real-world version of the environment has objects that were never seen during training (see Fig. 1b and 9). We evaluated 15 different models, trained to have varying degrees of robustness to

the *training and test domain gap*, based on domain randomization and randomized–to-canonical adaptation networks [12].[3] Out of these, 7 were trained on-policy purely in simulation. True average return was evaluated over 714 episodes with 7 different sets of objects, and true policy real-world performance ranged from 17% to 91%. The validation dataset consisted of 4,000 real-world episodes, 40% of which were successful grasps and the objects used for it were separate from the ones used for final evaluation used for the results in Table 2.

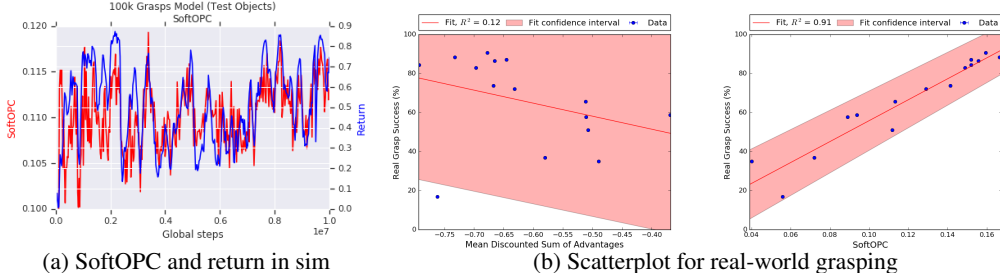

(a) SoftOPC and return in sim          (b) Scatterplot for real-world grasping

Figure 3: **(a): SoftOPC in simulated grasping.** Overlay of SoftOPC (red) and return (blue) in simulation for model trained with 100k grasps. SoftOPC tracks episode return. **(b): Scatterplots for OPE metrics and real-world grasp success.** Scatterplots for $\sum \gamma^{t'} A^\pi(\mathbf{s}_{t'}, \mathbf{a}_{t'})$ (left) and SoftOPC (right) for the Real-World grasping task. Each point is a different grasping model. Shaded regions are a 95% confidence interval. $\sum \gamma^{t'} A^\pi(\mathbf{s}_{t'}, \mathbf{a}_{t'})$ works in simulation but fails on real data, whereas SoftOPC works well in both.

## 6.3 Discussion

Table 2 shows $R^2$ and $\xi$ for each metric for the different environments we considered. Our proposed SoftOPC and OPC consistently outperformed the baselines, with the exception of the simulated robotic test environment, on which the SoftOPC performed almost as well as the discounted sum of advantages on the Spearman correlation (but worse on $R^2$). However, we show that SoftOPC more reliably ranks policies than the baselines for real-world performance without any real-world interaction, as one can also see in Fig. 3b. The same figure shows the sum of advantages metric that works well in simulation performs poorly in the real-world setting we care about. Appendix F includes additional experiments showing correlation mostly unchanged on different validation datasets.

Furthermore, we demonstrate that SoftOPC can track the performance of a policy acting in the simulated grasping environment, as it is training in Fig. 3a, which could potentially be useful for early stopping. Finally, SoftOPC seems to be performing slightly better than OPC in most of the experiments. We believe this occurs because the Q-functions compared in each experiment tend to have similar magnitudes. Preliminary results in Appendix H suggest that when Q-functions have different magnitudes, OPC might outperform SoftOPC.

## 7 Conclusion and future work

We proposed OPC and SoftOPC, classification-based off-policy evaluation metrics that can be used together with Q-learning algorithms. Our metrics can be used with binary reward tasks: tasks where each episode results in either a failure (zero return) or success (a return of one). While this class of tasks is a substantial restriction, many practical tasks actually fall into this category, including the real-world robotics tasks in our experiments. The analysis of these metrics shows that it can approximate the expected return in deterministic binary reward MDPs. Empirically, we find that OPC and the SoftOPC variant correlate well with performance across several environments, and predict generalization performance across several scenarios. including the simulation-to-reality scenario, a critical setting for robotics. Effective off-policy evaluation is critical for real-world reinforcement learning, where it provides an alternative to expensive real-world evaluations during algorithm development. Promising directions for future work include developing a variant of our method that is not restricted to binary reward tasks. We include some initial work in Appendix J. However, even in the binary setting, we believe that methods such as ours can provide for a substantially more practical pipeline for evaluating transfer learning and off-policy reinforcement learning algorithms.

**Acknowledgements**

We would like to thank Razvan Pascanu, Dale Schuurmans, George Tucker, and Paul Wohlhart for valuable discussions.

## Footnotes

[1] We slightly abuse notation here, and should clarify that $R^2$ is used to symbolize the coefficient of determination and should not be confused with $R(\pi)$, the average return of a policy $\pi$.

[2]Code for the binary tree environment is available at `https://bit.ly/2Qx6TJ7`.

[3]For full details of each of the models please see Appendix E.4.

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
