[Supplementary Material]

# Appendix for Off-Policy Evaluation via Off-Policy Classification

## A  Classification error bound

### A.1  Trajectory $\tau$ return $1 \Rightarrow$ all $\mathbf{a}_t$ feasible

Suppose this were not true. Then, there exists a time $t$ where $\mathbf{a}_t$ is catastrophic. After executing said $\mathbf{a}_t$, it should be impossible for $\tau$ to end in a success state. However, we know $\tau$ ends in success. By contradiction, all $(\mathbf{s}_t, \mathbf{a}_t)$ should be feasible.

### A.2  Proof of Theorem 1

To bound $R(\pi)$, it is easier to bound the failure rate $1 - R(\pi)$. Policy $\pi$ succeeds if and only if at every $\mathbf{s}_t$, it selects a feasible $\mathbf{a}_t$ a feasible $\mathbf{a}_t$, and dynamics do not take it to a catastrophic $\mathbf{s}_{t+1}$. The failure rate is the total probability $\pi$ makes its first mistake or is first brought to a catastrophic $\mathbf{s}_{t+1}$, summed over all $t$.

The distribution $\rho_{t,\pi}^+$ is defined such that $\rho_{t,\pi}^+$ is the marginal state distribution at time $t$, conditioned on $\mathbf{a}_1, \cdots, \mathbf{a}_{t-1}$ being feasible and $\mathbf{s}_t$ being feasible. This makes $\epsilon_t$ the expected probability $\pi$ choosing a catastrophic $\mathbf{a}_t$ at time $t$, given no mistakes earlier in the trajectory. Conditioned on this, the failure rate at time $t$ is upper-bounded by $\epsilon_t + c$.

$$1 - R(\pi) \leq \sum_{t=1}^{T} p(\pi \text{ at feasible } \mathbf{s}_t) \cdot (\epsilon_t + c) \tag{6}$$

$$\leq \sum_{t=1}^{T} (\epsilon_t + c) \tag{7}$$

$$\leq T(\epsilon + c) \tag{8}$$

This gives $R(\pi) \geq 1 - T(\epsilon + c)$ as desired. This bound is tight when $\pi$ is always at a feasible $\mathbf{s}_t$, which occurs when $c = 0$, $\epsilon_1 = \epsilon_2 = \cdots = \epsilon_{T-1} = 0$, and $\epsilon_T = T\epsilon$. When $c > 0$, this bound may be improvable.

### A.3  Alternate proof connecting to behavioral cloning in deterministic case

In a deterministic environment, we have $c(\mathbf{s}, \mathbf{a}) = 0$ for all $(\mathbf{s}, \mathbf{a})$, and a policy that only picks feasible actions will always achieve the optimal return of 1. Any such policy can be considered an expert policy $\pi^*$. Since $\epsilon$ is defined as the 0-1 loss over states conditioned on not selecting a catastrophic action, we can view $\epsilon$ as the 0-1 behavior cloning loss to an expert policy $\pi^*$. In this section, we present an alternate proof based on behavioral cloning bounds from Ross & Bagnell [32].

Theorem 2.1 of Ross & Bagnell [32] proves a $O(T^2\epsilon)$ cost bound for general MDPs. This differs from the $O(T\epsilon)$ cost derived above. The difference in bound comes because Ross & Bagnell [32] derive their proof in a general MDP, whose cost is upper bounded by 1 at every timestep. If $\pi$ deviates from the expert, it receives cost 1 several times, once for every future timestep. In binary reward MDPs, we only receive this cost once, at the final timestep. Transforming the proof to incorporate our binary reward MDP assumptions lets us recover the $O(T\epsilon)$ upper bound from Appendix A.2. We briefly explain the updated proof, using notation from [32] to make the connection more explicit.

Define $\epsilon_t$ as the expected 0-1 loss at time $t$ for $\pi$ under the state distribution of $\pi^*$. Since $\rho_{t,\pi}^+$ corresponds to states $\pi$ visits conditioned on never picking a catastrophic action, this is the same as our definition of $\epsilon_t$. The MDP is defined by cost instead of reward: cost $C(\mathbf{s})$ of state $\mathbf{s}$ is 0 for all timesteps except the final one, where $C(\mathbf{s}) \in \{0, 1\}$. Let $p_t$ be the probability $\pi$ hasn't made a mistake (w.r.t $\pi^*$) in the first $t$ steps, $d_t$ be the state distribution conditioned on no mistakes in the first $t-1$ steps, and $d_t'$ be the state distribution conditioned on $\pi$ making at least 1 mistake. In a general MDP with $0 \leq C(\mathbf{s}) \leq 1$, total cost $J(\pi)$ is bounded by $J(\pi) \leq \sum_{t=1}^{T} [p_{t-1}\mathbb{E}_{d_t}[C_\pi(\mathbf{s})] + (1 - p_{t-1})]$, where the 1st term is cost while following the expert and the 2nd term is an upper bound of cost 1 if

outside of the expert distribution. In a binary reward MDP, since $C(\mathbf{s}) = 0$ for all $t$ except $t = T$, we can ignore every term in the summation except the final one, giving

$$J(\pi) = p_{T-1}\mathbb{E}_{d_T}\left[C_\pi(\mathbf{s}_T)\right] + (1 - p_{T-1}) \tag{9}$$

Note $\mathbb{E}_{d_T}\left[C_\pi(\mathbf{s}_T)\right] = \epsilon_T$, and as shown in the original proof, $p_t \geq 1 - \sum_{i=1}^{t} \epsilon_i$. Since $p_{T-1}$ is a probability, we have $p_{T-1}\mathbb{E}_{d_T}\left[C_\pi(\mathbf{s}_T)\right] \leq \mathbb{E}_{d_T}\left[C_\pi(\mathbf{s}_T)\right]$, which recovers the $O(T\epsilon)$ bound, and again this is tight when $\epsilon_1 = \cdots = \epsilon_{T-1} = 0, \epsilon_T = T\epsilon$.

$$J(\pi) \leq \mathbb{E}_{d_T}\left[C_\pi(\mathbf{s}_T)\right] + \sum_{t=1}^{T-1}\epsilon_t = \sum_{t=1}^{T}\epsilon_T = T\epsilon \tag{10}$$

## B   Algorithm pseudocode

We first present pseudocode for SoftOPC.

$$\text{SoftOPC}(Q) = p(y = 1)\mathbb{E}_{(\mathbf{s},\mathbf{a}),y=1}\left[Q(\mathbf{s},\mathbf{a})\right] - \mathbb{E}_{(\mathbf{s},\mathbf{a})}\left[Q(\mathbf{s},\mathbf{a})\right]. \tag{11}$$

---

**Algorithm 1** Soft Off-Policy Classification (SoftOPC)

---

**Require:** Dataset $\mathcal{D}$ of trajectories $\tau = (\mathbf{s}_1, \mathbf{a}_1, \ldots, \mathbf{s}_T, \mathbf{a}_T)$, learned Q-function $Q(\mathbf{s}, \mathbf{a})$, prior $p(y = 1)$ (set to 1 for all experiments).
1: PositiveAverages $\leftarrow EMPTY\_LIST$
2: AllAverages $\leftarrow EMPTY\_LIST$
3: **for** $(\mathbf{s}_1, \mathbf{a}_1, \ldots, \mathbf{s}_T, \mathbf{a}_T, r_T) \in \mathcal{D}$ **do**
4:     Compute Q-values $Q(\mathbf{s}_1, \mathbf{a}_1), \ldots, Q(\mathbf{s}_T, \mathbf{a}_T)$.
5:     AverageQ $\leftarrow \frac{1}{T}\sum_t Q(\mathbf{s}_t, \mathbf{a}_t)$
6:     AllAverages.append(AverageQ)
7:     **if** $r_T = 1$ **then**
8:         PositiveAverages.append(AverageQ)
9:     **end if**
10: **end for**
11: **return** $p(y = 1)AVERAGE(\text{PositiveAverages}) - AVERAGE(\text{AllAverages})$

---

Next, we present pseudocode for OPC.

$$\text{OPC}(Q) = p(y = 1)\mathbb{E}_{(\mathbf{s},\mathbf{a}),y=1}\left[1_{Q(s,a)>b}\right] - \mathbb{E}_{(\mathbf{s},\mathbf{a})}\left[1_{Q(s,a)>b}\right] \tag{12}$$

Suppose we have $N$ transitions, of which $N^+$ of them have positive labels. Imagine placing each $Q(\mathbf{s}, \mathbf{a})$ on the number line. Each $Q(\mathbf{s}, \mathbf{a})$ is annotated with a score, $-1/N$ for unlabeled transitions and $p(y = 1)/N^+ - 1/N$ for positive labeled transitions. Imagine sliding a line from $b = -\infty$ to $b = \infty$. At $b = -\infty$, the OPC score is $p(y = 1) - 1$. The OPC score only updates when this line passes some $Q(\mathbf{s}, \mathbf{a})$ in our dataset, and is updated based on the score annotated at $Q(\mathbf{s}, \mathbf{a})$. After sorting by Q-value, we can find all possible OPC scores in $O(N)$ time by moving $b$ from $-\infty$ to $\infty$, noting the updated score after each $Q(\mathbf{s}, \mathbf{a})$ we pass. Given $\mathcal{D}$, we sort the $N$ transitions in $O(N \log N)$, annotate them appropriately, then compute the maximum over all OPC scores. When doing so, we must be careful to detect when the same $Q(\mathbf{s}, \mathbf{a})$ value appears multiple times, which can occur when $(\mathbf{s}, \mathbf{a})$ appears in multiple trajectories.

## C   Baseline metrics

We elaborate on the exact expressions used for the baseline metrics. In all baselines, $\mathbf{a}_t^\pi$ is the on-policy action $\arg\max_{\mathbf{a}} Q^\pi(\mathbf{s}_t, \mathbf{a})$.

**Temporal-difference error**   The TD error is the squared error between $Q(\mathbf{s}, \mathbf{a})$ and the 1-step return estimate of the action's value.

$$\mathbb{E}_{\mathbf{s}_t,\mathbf{a}_t \sim \pi_b}\left[(Q^\pi(\mathbf{s}_t, \mathbf{a}_t) - (r_t + \gamma Q^\pi(\mathbf{s}_{t+1}, \mathbf{a}_t^\pi)))^2\right] \tag{13}$$

---

**Algorithm 2** Off-Policy Classification (OPC)

---

**Require:** Dataset $\mathcal{D}$ of trajectories $\tau = (\mathbf{s}_1, \mathbf{a}_1, \ldots, \mathbf{s}_T, \mathbf{a}_T)$, learned Q-function $Q(\mathbf{s}, \mathbf{a})$, prior $p(y = 1)$ (set to 1 for all experiments).
1: $N^+ \leftarrow$ number of $(\mathbf{s}, \mathbf{a}) \in \mathcal{D}$ from trajectories where $r_T = 1$
2: $N \leftarrow$ number of $(\mathbf{s}, \mathbf{a}) \in \mathcal{D}$
3: Q-values $\leftarrow EMPTY\_DICTIONARY$
4: **for** $(\mathbf{s}_1, \mathbf{a}_1, \ldots, \mathbf{s}_T, \mathbf{a}_T, r_T) \in \mathcal{D}$ **do**        ▷ Prepare annotated number line
5:     Compute Q-values $Q(\mathbf{s}_1, \mathbf{a}_1), \ldots, Q(\mathbf{s}_T, \mathbf{a}_T)$.
6:     **for** $t = 1, 2, \ldots, T$ **do**
7:         **if** $Q(\mathbf{s}, \mathbf{a}) \notin$ Q-values.keys() **then**
8:             Q-values[$Q(\mathbf{s}, \mathbf{a})$] = 0
9:         **end if**
10:        **if** $r_T = 1$ **then**
11:            Q-values[$Q(\mathbf{s}, \mathbf{a})$] += $p(y = 1)/N^+$
12:        **else**
13:            Q-values[$Q(\mathbf{s}, \mathbf{a})$] += $-1/N$
14:        **end if**
15:     **end for**
16: **end for**
17: RunningTotal $\leftarrow p(y = 1) - 1$        ▷ OPC when $b = -\infty$
18: BestOPC $\leftarrow$ RunningTotal
19: **for** $b \in$ Sorted(Q-values.keys()) **do**
20:     RunningTotal += Q-values[$b$]
21:     BestOPC $\leftarrow$ max(BestOPC, RunningTotal)
22: **end for**
23: **return** BestOPC

---

**Discounted sum of advantages**    The difference of the value functions of two policies $\pi$ and $\pi_b$ at state $\mathbf{s}_t$ is given by the discounted sum of advantages [14, 26] of $\pi$ on episodes induced by $\pi_b$:

$$V^{\pi_b}(\mathbf{s}_t) - V^\pi(\mathbf{s}_t) = \mathbb{E}_{\mathbf{s}_t, \mathbf{a}_t \sim \pi_b} \left[ \sum_{t'=t}^T \gamma^{t'-t} A^\pi(\mathbf{s}_{t'}, \mathbf{a}_{t'}) \right], \tag{14}$$

where $\gamma$ is the discount factor and $A^\pi$ the advantage function for policy $\pi$, defined as $A^\pi(\mathbf{s}_t, \mathbf{a_t}) = Q^\pi(\mathbf{s}_t, \mathbf{a}_t) - Q^\pi(\mathbf{s}_t, \mathbf{a}_t^\pi)$. Since $V^{\pi_b}$ is fixed, estimating (14) is sufficient to compare $\pi_1$ and $\pi_2$. The $\pi$ with smaller score is better.

$$\mathbb{E}_{\mathbf{s}_t, \mathbf{a}_t \sim \pi_b} \left[ \sum_{t'=t}^T \gamma^{t'-t} A^\pi(\mathbf{s}_{t'}, \mathbf{a}_{t'}) \right]. \tag{15}$$

**Monte-Carlo estimate corrected with the discounted sum of advantages**    Estimating $V^{\pi_b}(\mathbf{s}_t) = \mathbb{E}_{\pi_b}[\sum_{t'} \gamma^{t'-t} r_{t'}]$ with the Monte-Carlo return, substituting into Eqn. (14), and rearranging gives

$$V^\pi(\mathbf{s}_t) = \mathbb{E}_{\mathbf{s}_t, \mathbf{a}_t \sim \pi_b} \left[ \sum_{t'=t}^T \gamma^{t'-t} \left( r_{t'} - A^\pi(\mathbf{s}_{t'}, \mathbf{a}_{t'}) \right) \right] \tag{16}$$

With $V^\pi(\mathbf{s}_t) + A^\pi(\mathbf{s}_t, \mathbf{a}_t) = Q^\pi(\mathbf{s}_t, \mathbf{a}_t)$, we can obtain an approximate $\widetilde{Q}$ estimate depending on the whole episode:

$$\widetilde{Q}_{MCC}(\mathbf{s}_t, \mathbf{a}_t, \pi) = \mathbb{E}_{\mathbf{s}_t, \mathbf{a}_t \sim \pi_b} \left[ r_t + \sum_{t'=t+1}^T \gamma^{t'-t} (r_{t'} - A^\pi(\mathbf{s}_{t'}, \mathbf{a}_{t'})) \right] \tag{17}$$

The MCC Error is the squared error to this estimate.

$$\mathbb{E}_{\mathbf{s}_t, \mathbf{a}_t \sim \pi_b} \left[ \left( Q^\pi(\mathbf{s}_t, \mathbf{a}_t) - \widetilde{Q}_{MCC}(\mathbf{s}_t, \mathbf{a}_t, \pi) \right)^2 \right] \tag{18}$$

Note that (18) was proposed before by Quillen et al. [29] as a training loss for a Q-learning variant, but not for the purpose of off-policy evaluation.

Eqn. (13) and Eqn. (17) share the same optimal Q-function, so assuming a perfect learning algorithm, there is no difference in information between these metrics. In practice, the Q-function will not be perfect due to errors in function approximation and optimization. Eqn. (17) is designed to rely on all future rewards from time $t$, rather than just $r_t$. We theorized that using more of the "ground truth" from $\mathcal{D}$ could improve the metric's performance in imperfect learning scenarios. This did not occur in our experiments - the MCC Error performed poorly.

## D   Argument for choosing $p(y = 1) = 1$

The positive class prior $p(y = 1)$ should intuitively depend on the environment, since some environments will have many more feasible $(\mathbf{s}, \mathbf{a})$ than others. However, recall how error $\epsilon$ is defined. Each $\epsilon_t$ is defined as:

$$\epsilon_t = \mathbb{E}_{\rho_{t,\pi}^+} \left[ \sum_{\mathbf{a} \in \mathcal{A}_-(\mathbf{s}_t)} \pi(\mathbf{a}|\mathbf{s}_t) \right] \tag{19}$$

where state distribution $\rho_{t,\pi}^+$ is defined such that $\mathbf{a}_1, \cdots, \mathbf{a}_{t-1}$ were all feasible. This is equivalent to following an optimal "expert" policy $\pi^*$, and although we are estimating $\epsilon_t$ from data generated by behavior policy $\pi_b$, we should match the positive class prior $p(y = 1)$ we would observe from expert $\pi^*$. Assuming the task is feasible, meaining the policy has feasible actions available from the start, we have $R(\pi^*) = 1$. Therefore, although the validation dataset will likely have both successes and failures, a prior of $p(y = 1) = 1$ is the ideal prior, and this holds independently of the environment. As a didactic toy example, we show this holds for a binary tree domain. In this domain, each node is a state, actions are $\{left, right\}$, and leaf nodes are terminal with reward 0 or 1. We try $p(y = 1) \in \{0, 0.05, 0.1, \cdots, 0.9, 0.95, 1\}$ in two extremes: only 1 leaf fails, or only 1 leaf succeeds. No stochasticity was added. Validation data is from the uniformly random policy. The frequency of feasible $(\mathbf{s}, \mathbf{a})$ varies a lot between the two extremes, but in both Spearman correlation $\rho$ monotonically increases with $p(y = 1)$ and was best with $p(y = 1) = 1$. Fig. 4 shows Spearman correlation of OPC and SoftOPC with respect to $p(y = 1)$, when the tree is mostly success or failures. In both settings $p(y = 1) = 1$ has the best correlation.

Figure 4: Spearman correlation of SoftOPC, OPC, and baselines with varying $p(y = 1)$. Baselines do not depend on $p(y = 1)$. Correlations further from 0 are better.

From an implementation perspective, $p(y = 1) = 1$ is also the only choice that can be applied across arbitrary validation datasets. Suppose $\pi_b$, the policy collecting our validation set, succeeds with probability $R(\pi_b) = p$. In the practical computation of OPC presented in Algorithm 2, we have $N$ transitions, and $pN$ of them have positive labels. Each $Q(\mathbf{s}, \mathbf{a})$ is annotated with a score: $\frac{-1}{N}$ for unlabeled transitions and $\frac{p(y=1)}{pN} - \frac{1}{N}$ for positive labeled transitions. The maximal OPC score will be the sum of all annotations within the interval $[b, \infty)$, and we maximize over $b$.

For unlabeled transitions, the annotation is $-1/N$, which is negative. Suppose the annotation for positive transitions was negative as well. This occurs when $p(y = 1)/(pN) - 1/N < 0$. If every

annotation is negative, then the optimal choice for $b$ is $b = \infty$, since the empty set has total 0 and every non-empty subset has a negative total. This gives $\text{OPC}(Q) = 0$, no matter what $Q(\mathbf{s}, \mathbf{a})$ we are evaluating, which makes the OPC score entirely independent of episode return.

This degenerate case is undesirable, and happens when $p(y = 1)/(pN) < 1/N$, or equivalently $p(y = 1) < p$. To avoid this, we should have $p(y = 1) \geq p = R(\pi_b)$. If we wish to pick a single $p(y = 1)$ that can be applied to data from arbitrary behavior policies $\pi_b$, then we should pick $p(y = 1) \geq R(\pi^*)$. In binary reward MDPs where $\pi^*$ can always succeed, this gives $p(y = 1) \geq R(\pi^*) = 1$, and since the prior is a probability, it should satisfy $0 \leq p(y = 1) \leq 1$, leaving $p(y = 1) = 1$ as the only option.

To complete the argument, we must handle the case where we have a binary reward MDP where $R(\pi^*) < 1$. In a binary reward MDP, the only way to have $R(\pi^*) < 1$ is if the sampled initial state $\mathbf{s}_1$ is one where $(\mathbf{s}_1, \mathbf{a})$ is catastrophic for all $\mathbf{a}$. From these $\mathbf{s}_1$, and all future $\mathbf{s}_t$ reachable from $\mathbf{s}_1$, the actions $\pi$ chooses do not matter - the final return will always be 0. Since $\epsilon_t$ is defined conditioned on only executing feasible actions so far, it is reasonable to assume we only wish to compute the expectation over states where our actions can impact reward. If we computed optimal policy return $R(\pi^*)$ over just the initial states $\mathbf{s}_1$ where feasible actions exist, we have $R(\pi^*) = 1$, giving $1 \leq p(y = 1) \leq 1$ once again.

# E   Experiment details

## E.1   Binary tree environment details

The binary tree is a full binary tree with $k = 6$ levels. The initial state distribution is uniform over all non-leaf nodes. Initial state may sometimes be initialized to one where failure is inevitable. The validation dataset is collected by generating 1,000 episodes from the uniformly random policy. For Q-functions, we generate 1,000 random Q-functions by sampling $Q(\mathbf{s}, \mathbf{a}) \sim U[0, 1]$ for every $(\mathbf{s}, \mathbf{a})$, defining the policy as $\pi(\mathbf{s}) = \arg\max_{\mathbf{a}} Q(\mathbf{s}, \mathbf{a})$. We try priors $p(y = 1) \in \{0, 0.05, 0.1, \cdots, 0.9, 0.95, 1\}$. Code for this environment is available at https://gist.github.com/alexirpan/54ac855db7e0d017656645ef1475ac08.

## E.2   Pong details

Fig. 5 is a scatterplot of our Pong results. Each color represents a different hyperparameter setting, as explained in the legend. From top to bottom, the abbreviations in the legend mean:

- `DQN`: trained with DQN
- `DDQN`: trained with Double DQN
- `DQN_gamma9`: trained with DQN, $\gamma = 0.9$ (default is 0.99).
- `DQN2`: trained with DQN, using a different random seed
- `DDQN2`: trained with Double DQN, using a different random seed
- `DQN_lr1e4`: trained with DQN, learning rate $10^{-4}$ (default is $2.5 \times 10^{-4}$).
- `DQN_b64`: trained with DQN, batch size 64 (default is 32).
- `DQN_fixranddata`: The replay buffer is filled entirely by a random policy, then a DQN model is trained against that buffer, without pushing any new experience into the buffer.
- `DDQN_fixranddata`: The replay buffer is filled entirely by a random policy, then a Double DQN model is trained against that buffer, without pushing new experience into the buffer.

In Fig. 5, models trained with $\gamma = 0.9$ are highlighted. We noticed that SoftOPC was worse at separating these models than OPC, suggesting the 0-1 loss is preferable in some cases. This is discussed further in Appendix H.

In our implementation, all models were trained in the full version of the Pong game, where the maximum return possible is 21 points. However, to test our approach we create a binary version for evaluation. Episodes in the validation set were truncated after the first point was scored. Return of the policy was estimated similarly: the policy was executed until the first point is scored, and the average

return is computed over these episodes. Although the train time environment is slightly different from the evaluation environment, this procedure is fine for our method, since our method can handle environment shift and we treat $Q(\mathbf{s}, \mathbf{a})$ as a black-box scoring function. OPC can be applied as long as the validation dataset matches the semantics of the test environment where we estimate the final return.

Figure 5: Scatterplot of episode return (x-axis) of Pong models against metric (y-axis), for SoftOPC (left) and OPC (right). Each color is a set of model checkpoints from a different hyperparameter setting, with the legend explaining the mapping from color to hyperparameters. In each plot, points trained with DQN, $\gamma = 0.9$ are boxed with a red rectangle. We observed that the hard 0-1 loss in OPC does a better job separating these models than the soft loss in SoftOPC.

## E.3 Simulated grasping details

The objects we use were generated randomly through procedural generation. The resulting objects are highly irregular and are often non-convex. Some example objects are shown in Fig. 6a.

(a) Procedurally generated objects for simulation   (b) Test objects for real-world evaluation

Figure 6: **(a):** Example objects from the procedural generation process used during training in simulation. **(b):** Real test objects used during evaluation on real robots.

Fig. 7 demonstrates two generalization problems from Sect. 4: *insufficient off-policy training data* and *mismatched off-policy training data*. We trained two models with a limited 100k grasps dataset or a large 900k grasps dataset, then evaluated grasp success. The model with limited data fails to achieve stable grasp success due to overfitting to its limited dataset. Meanwhile, the model with abundant data learns to model the train objects, but fails to model the test objects, since they are unobserved at training time. The train and test objects used are show in Fig. 8.

## E.4 Real-world grasping

Several visual differences between simulation and reality limit the real performance of model trained in simulation (see Fig. 9) and motivate simulation-to-reality methods such as the Randomized-to-Canonical Adaptation Networks (RCANs), as proposed by James et al. [12]. The 15 real-world grasping models evaluated were trained using variants of RCAN. These networks train a generator to

Figure 7: *Left:* Grasp success curve of model trained with 900k or 100k grasps. The 100k grasps model oscillates in performance. *Middle:* We see why: holdout TD error (blue) of the 100k grasps model is increasing. *Right*: The TD Error for the 900k grasp model is the same for train and holdout, but is still larger for test data on unseen test objects.

(a) Train objects            (b) Test objects

Figure 8: **Example of mismatched off-policy training data.** Train objects and test objects from simulated grasping task in Sect. 6.2. Given large amounts of data from train objects, models do not fully generalize to test objects.

transform randomized simulation images to a canonical simulated image. A policy is learned over this canonical simulated image. At test time, the generator transforms real images to the same canonical simulated image, facilitating zero-shot transfer. Optionally, the policy can be fine-tuned with real-world data, in this case the real-world training objects are distinct from the evaluation objects. The SoftOPC and real-world grasp success of each model is listed in Table 3. From top-to-bottom, the abbreviations mean:

- `Sim`: A model trained only in simulation.

- `Randomized Sim`: A model trained only in simulation with the *mild randomization* scheme from James et al. [12]: random tray texture, object texture and color, robot arm color, lighting direction and brightness, and one of 6 background images consisting of 6 different images from the view of the real-world camera.

- `Heavy Randomized Sim`: A model trained only in simulation with the *heavy randomization* scheme from James et al. [12]: every randomization from *Randomized Sim*, as well as slight randomization of the position of the robot arm and tray, randomized position of the divider within the tray (see Figure 1b in main text for a visual of the divider), and a more diverse set of background images.

- `Randomized Sim + Real (2k)`: The `Randomized Sim` Model, after training on an additional 2k grasps collected on-policy on the real robot.

- `Randomized Sim + Real (3k)`: The `Randomized Sim` Model, after training on an additional 3k grasps collected on-policy on the real robot.

- `Randomized Sim + Real (4k)`: The `Randomized Sim` Model, after training on an additional 4k grasps collected on-policy on the real robot.

- `Randomized Sim + Real (5k)`: The `Randomized Sim` Model, after training on an additional 5k grasps collected on-policy on the real robot.

- `RCAN`: The RCAN model, as described in [12], trained in simulation with a pixel level adaptation model.

Figure 9: Visual differences for the robot grasping task between simulation (left) and reality (right). In simulation, models are trained to grasp procedurally generated shapes while in reality objects have more complex shapes. The simulated robot arm does not have the same colors and textures as the real robot and lacks the visible real cables. The tray in reality has a greater variation in appearance than in the simulation.

- `RCAN + Real (2k)`: The `RCAN` model, after training on an additional 2k grasps collected on-policy on the real robot.
- `RCAN + Real (3k)`: The `RCAN` model, after training on an additional 3k grasps collected on-policy on the real robot.
- `RCAN + Real (4k)`: The `RCAN` model, after training on an additional 4k grasps collected on-policy on the real robot.
- `RCAN + Real (5k)`: The `RCAN` model, after training on an additional 5k grasps collected on-policy on the real robot.
- `RCAN + Dropout`: The `RCAN` model with dropout applied in the policy.
- `RCAN + InputDropout`: The `RCAN` model with dropout applied in the policy and RCAN generator.
- `RCAN + GradsToGenerator`: The `RCAN` model where the policy and RCAN generator are trained simultaneously, rather than training RCAN first and the policy second.

## F SoftOPC performance on different validation datasets

For real grasp success we use 7 KUKA LBR IIWA robot arms to each make 102 grasp attempts from 7 different bins with test objects (see Fig. 6b). Each grasp attempt is allowed up to 20 steps and any grasped object is dropped back in the bin, a successful grasp is made if any of the test objects is held in the gripper at the end of the episode.

For estimating SoftOPC, we use a validation dataset collected from two policies, a poor policy with a success of 28%, and a better policy with a success of 51%. We divided the validation dataset based on the policy used, then evaluated SoftOPC on data from only the poor or good policy. Fig. 10 shows the correlation on these subsets of the validation dataset. The correlation is slightly worse on the poor dataset, but the relationship between SoftOPC and episode reward is still clear.

As an extreme test of robustness, we go back to the simulated grasping environment. We collect a new validation dataset, using the same human-designed policy with $\epsilon = 0.9$ greedy exploration instead. The resulting dataset is almost all failures, with only 1% of grasps succeeding. However, this dataset also covers a broad range of states, due to being very random. Fig. 11 shows the OPC

| Model | SoftOPC | Real Grasp Success (%) |
|---|---|---|
| Sim | 0.056 | 16.67 |
| Randomized Sim | 0.072 | 36.92 |
| Heavy Randomized Sim | 0.040 | 34.90 |
| Randomized Sim + Real (2k) | 0.129 | 72.14 |
| Randomized Sim + Real (3k) | 0.141 | 73.65 |
| Randomized Sim + Real (4k) | 0.149 | 82.92 |
| Randomized Sim + Real (5k) | 0.152 | 84.38 |
| RCAN | 0.113 | 65.69 |
| RCAN + Real (2k) | 0.156 | 86.46 |
| RCAN + Real (3k) | 0.166 | 88.34 |
| RCAN + Real (4k) | 0.152 | 87.08 |
| RCAN + Real (5k) | 0.159 | 90.71 |
| RCAN + Dropout | 0.112 | 51.04 |
| RCAN + InputDropout | 0.089 | 57.71 |
| RCAN + GradsToGenerator | 0.094 | 58.75 |

Table 3: Real-world grasping models used for Sect. 6.2 simulation-to-reality experiments.

Figure 10: **SoftOPC versus the real grasp success over different validation datasets for Real-World grasping.** *Left*: SoftOPC over entire validation dataset. *Middle*: SoftOPC over validation data from only the poor policy (28% success rate). *Right*: SoftOPC over validation data from only the better policy (51% success). In each, a fitted regression line with its $R^2$ and 95% confidence interval is also shown.

and SoftOPC still perform reasonably well, despite having very few positive labels. From a practical standpoint, this suggests that OPC or SoftOPC have some robustness to the choice of generation process for the validation dataset.

Figure 11: **SoftOPC and OPC over almost random validation data on test objects in simulated grasping.** We generate a validation dataset from an $\epsilon$-greedy policy where $\epsilon = 0.9$, leading to a validation dataset where only 1% of episodes succeed. *Left*: SoftOPC over the poor validation dataset. $R^2 = 0.83, \rho = 0.94$. *Right*: OPC over the poor validation dataset. $R^2 = 0.83, \rho = 0.88$.

# G  Plots of Q-value distributions

In Fig. 12, we plot the Q-values of two real-world grasping models. The first is trained only in simulation and has poor real-world grasp success. The second is trained with a mix of simulated and real-world data. We plot a histogram of the average Q-value over each episode of validation set $\mathcal{D}$. The better model has a wider separation between successful Q-values and failed Q-values.

Figure 12: **Q-value distributions for successful and failed episodes.** *Left:* Q-value distributions over successful and failed episodes in an off-policy data-set according to a learned policy with a poor grasp success rate of 36%. *Right:* The same distributions after the learned policy is improved by 5,000 grasps of real robot data, achieving a 84% grasp success rate.

(a) $Q(\mathbf{s}, \mathbf{a}) \sim U[0, k]$            (b) $Q(\mathbf{s}, \mathbf{a}) \sim U[0, 1000]$

Figure 13: Spearman correlation in binary tree with one success state for different Q-function generation methods. Varying magnitudes between Q-functions causes the SoftOPC to perform worse.

# H  Comparison of OPC and SoftOPC

We elaborate on the argument presented in the main paper, that OPC performs better when $Q(\mathbf{s}, \mathbf{a})$ have different magnitudes, and otherwise SoftOPC does better. To do so, it is important to consider how the Q-functions were trained. In the tree environments, $Q(\mathbf{s}, \mathbf{a})$ was sampled uniformly from $U[0, 1]$, so $Q(\mathbf{s}, \mathbf{a}) \in [0, 1]$ by construction. In the grasping environments, the network architecture ends in $\mathrm{sigmoid}(x)$, so $Q(\mathbf{s}, \mathbf{a}) \in [0, 1]$. In these experiments, SoftOPC did better. In Pong, $Q(\mathbf{s}, \mathbf{a})$ was not constrained in any way, and these were the only experiments where discount factor $\gamma$ was varied between models. Here, OPC did better.

The hypothesis that Q-functions of varying magnitudes favor OPC can be validated in the tree environment. Again, we evaluate 1,000 Q-functions, but instead of sampling $Q(\mathbf{s}, \mathbf{a}) \sim U[0, 1]$, the $k$th Q-function is sampled from $Q(\mathbf{s}, \mathbf{a}) \sim U[0, k]$. This produces 1,000 different magnitudes between the compared Q-functions. Fig. 13a demonstrates that when magnitudes are deliberately changed for each Q-function, the SoftOPC performs worse, whereas the non-parametric OPC is

unchanged. To demonstrate this is caused by a difference in magnitude, rather than large absolute magnitude, OPC and SoftOPC are also evaluated over $Q(\mathbf{s}, \mathbf{a}) \sim U[0, 1000]$. Every Q-function has high magnitude, but their magnitudes are consistently high. As seen in Fig. 13b, in this setting the SoftOPC goes back to outperforming OPC.

# I   Scatterplots of each metric

In Figure 14, we present scatterplots of each of the metrics in the simulated grasping environment from Sect. 6.2. We trained two Q-functions in a fully off-policy fashion, one with a dataset of $100,000$ episodes, and the other with a dataset of $900,000$ episodes. For every metric, we generate a scatterplot of all the model checkpoints. Each model checkpoint is color coded by whether it was trained with $100,000$ episodes or $900,000$ episodes.

Figure 14: **Scatterplots of each metric in simulated grasping over train objects.** From left-to-right, top-to-bottom, we present scatterplots for: the TD error, $\sum \gamma^{t'} A^\pi(\mathbf{s}_{t'}, \mathbf{a}_{t'})$, MCC error, OPC, and SoftOPC.

# J   Extension to non-binary reward tasks

Here, we present a template for how we could potentially extend OPC to non-binary, dense rewards.

First, we reduce dense reward MDPs to a sparse reward MDP, by applying a return-equivalent reduction introduced by Arjona-Medina et al. [1]. Given two MDPs, the two are *return-equivalent* if

(i) they differ only in the reward distribution and (ii) they have the same expected return at $t = 0$ for every policy $\pi$. We show all dense reward MDPs can be reduced to a return-equivalent sparse reward MDP.

Given any MDP $(\mathcal{S}, \mathcal{A}, \mathcal{P}, \mathcal{S}_0, r, \gamma)$, we can augment the state space to create a return-equivalent MDP. First, augment state $\mathbf{s}$ to be $(\mathbf{s}, \mathbf{r})$, where $\mathbf{r}$ is an added feature for accumulating reward. At the initial state, $\mathbf{r} = 0$. At time $t$ in the original MDP, whenever the policy would receive reward $r_t$, instead it receives no reward, but $r_t$ is added to the $\mathbf{r}$ feature. This gives $(\mathbf{s}_1, 0), (\mathbf{s}_2, r_1), (\mathbf{s}_3, r_1 + r_2)$, and so on. Adding $\mathbf{r}$ lets us maintain the Markov property, and at the final timestep, the policy receives reward equal to accumulated reward $\mathbf{r}$.

This MDP is return-equivalent and a sparse reward task. However, we do note that this requires adding an additional feature to the state space. Q-functions $Q(\mathbf{s}, \mathbf{a})$ trained in the original MDP will not be aware of $\mathbf{r}$. To handle this, note that for $Q(\mathbf{s}, \mathbf{a})$, the equivalent Q-function $Q'(\mathbf{s}, \mathbf{r}, \mathbf{a})$ in the new MDP should satisfy $Q'(\mathbf{s}, \mathbf{r}, \mathbf{a}) = \mathbf{r} + Q(\mathbf{s}, \mathbf{a})$, since $\mathbf{r}$ is return so far and $Q(\mathbf{s}, \mathbf{a})$ is estimated future return from the original MDP. At evaluation time, we can compute $\mathbf{r}$ at each $t$ and adjust Q-values accordingly.

This reduction lets us consider just the sparse non-binary reward case. To do so, we first use a well-known lemma.

**Lemma 1.** *Let $X$ be a discrete random variable over the real numbers with $n$ outcomes, where $X \in \{c_1, c_2, \cdots, c_n\}$, outcome $c_i$ occurs with probability $p_i$, and without loss of generalization $c_1 < c_2 < \cdots < c_n$. Then $\mathbb{E}[X] = c_1 + \sum_{i=2}^{n}(c_i - c_{i-1})P(X \geq c_i)$.*

*Proof.* Consider the right hand side. Substituting $1 = \sum_i p_i$ and expanding $P(X \geq c_i)$ gives

$$c_1 \sum_{j=1}^{n} p_j + \sum_{i=2}^{n}(c_i - c_{i-1}) \sum_{j=i}^{n} p_j \tag{20}$$

For each $p_j$, the coefficient is $c_1 + (c_2 - c_1) + \cdots + (c_j - c_{j-1}) = c_j$. The total sum is $\sum_{j=1}^{n} c_j p_j$, which matches the expectation $\mathbb{E}[X]$. $\square$

The return $R(\pi)$ can be viewed as a random variable depending on $\pi$ and the environment. We add a simplifying assumption: the environment has a finite number of return outcomes, all of which appear in dataset $\mathcal{D}$. Letting those outcomes be $c_1, c_2, \cdots, c_n$, estimating $P(R(\pi) \geq c_i)$ for each $c_i$ would be sufficient to estimate expected return.

The key insight is that to estimate $P(R(\pi) \geq c_i)$, we can define one more binary reward MDP. In this MDP, reward is sparse, and the final return is 1 if original return is $\geq c_i$, and 0 otherwise. The return of $\pi$ in this new MDP is exactly $P(R(\pi) \geq c_i)$, and can be estimated with OPC. The final pseudocode is provided below.

---

**Algorithm 3** Thresholded Off-Policy Classification

---

**Require:** Dataset $\mathcal{D}$ of trajectories $\tau = (\mathbf{s}_1, \mathbf{a}_1, \ldots, \mathbf{s}_T, \mathbf{a}_T)$, learned Q-function $Q(\mathbf{s}, \mathbf{a})$, return threshold $c_i$
1: **for** $\tau \in \mathcal{D}$ **do**  ▷ Generate Q-values and check threshold
2:     **for** $t = 1, 2, \ldots, T$ **do**
3:         $\mathbf{r}_t \leftarrow \sum_{t'=1}^{t} r_{t'}$
4:         Compute $Q'(\mathbf{s}_t, \mathbf{r}, \mathbf{a}_t) = \mathbf{r}_t + Q(\mathbf{s}, \mathbf{a})$
5:     **end for**
6:     Save each $Q'(\mathbf{s}_t, \mathbf{r}, \mathbf{a}_t)$, as well as whether $\sum_{t=1}^{T} r_t$ exceeds $c_i$.
7: **end for**
8: **return** OPC for the computed $Q'(\mathbf{s}, \mathbf{r}, \mathbf{a})$ and threshold $c_i$.

---

---

**Algorithm 4** Extended Off-Policy Classification

---

**Require:** Dataset $\mathcal{D}$ of trajectories $\tau = (\mathbf{s}_1, \mathbf{a}_1, \ldots, \mathbf{s}_T, \mathbf{a}_T)$, learned Q-function $Q(\mathbf{s}, \mathbf{a})$.
  1: $c_1, \cdots, c_n \leftarrow$ The $n$ distinct return outcomes within $\mathcal{D}$, sorted
  2: total $\leftarrow c_1$
  3: **for** $i = 2, \ldots, n$ **do**
  4:     total $+= (c_i - c_{i-1})\text{ThresholdedOPC}(Q(\mathbf{s}, \mathbf{a}), \mathcal{D}, c_i)$
  5: **end for**
  6: **return** total

---