[Reviews · NeurIPS 2019]

Reviewer 1



In summary, I like the rationale for doing this but the problem domain is so restricted its hard for me to see the path forward to more realistic settings, with stochastic dynamics and dense rewards. Each of those two seems to break the algorithm by itself. A smaller comment: It is common to present "the algorithm" in one place as pseudocode. This paper doesn't have it and I think it would help the presentation.

Reviewer 2



Originality: As stated in the contributions, the paper presents an approach to OPE that is a departure from classic approaches. As far as I am aware this is a very novel way to address this issue Clarity: I found the paper to somewhat lack structure, which hurts readability and clarity: - The description of the positive unlabeled learning and its application to OPE (section 3) is rather convoluted and not always easy to follow. This section could use some intuitive examples and a more detailed step-by-step development. - Overall structure: Section 3 ends with short disconnected section 3.3 on metrics, then moves on to a largely unrelated section 4 motivating applicability, before section 5 discusses related work and more metrics. Significance & Quality: The paper addresses an important problem using a novel approach. It will hopefully inspire new approaches to the problem. The method as presented here do have some important limitations: - The method is restricted to deterministic systems with a single binary success reward per episode. As noted by the authors this setting is quite restrictive. Even though many problems may be interpreted as binary success problems, the inability to deal with stochasticity in the problem may have a large impact on applicability. It would also be good to test how important deterministic dynamics are to the performance of the method in practice. - The approach seems to simply ignore one of the key issues in OPE, i.e. the distribution mismatch in the data. In section 3.1, the authors note that they simply assume that the classification error is a good enough proxy and that it seems to be robust in their evaluations. This assumption is not sufficiently tested. It would be good to explicitly evaluate robustness in the Tree example and explicitly evaluate how / if the method degrades when the data is generated by policies that are not very broad/random or far away from the target policy. - Unlike existing OPE approaches, the method does not seem to offer strong theoretical guarantees / bounds on performance. - It was not clear to me how data efficient the method is. Lack of data is a key issue in the settings considered in the paper (i.e. where no simulation for the real test problem is possible), so it would be good to see how the performance depends on the amount of data.

Reviewer 3



Originality: The work seems novel, introducing both a new metric for OPPE, and a new way to benchmark deep RL algorithms Quality: Given the assumptions provided, the proofs all seem correct, and the method seems to work in the settings advertised. Clarity: The proofs and arguments are all easy to follow. It was hard for me to understand how the correlation coefficient, and I hope I reached the right understanding. Perhaps a more detailed series of steps as I outline below would help clarify the steps needed. Significance: If this is more generalizable to less restrictive assumptions, it could be a very novel way to do OPPE, and might advance the whole domain. L72: Why does the \gamma = 1 restriction exist? It doesn't seem to be used in the proofs, and the experiments seem to have bene done with \gamma != 1. The assumption that the transition matrix P being deterministic does not seem like a good assumption for many real world problems, although it's likely not a bad one for the specific robot grasping task. For example: due to imprecision in hardware, the same motor command in the same state twice will probably lead to a set of states. Specifically in the tasks chosen, there is very little stochasticity - the robotic hand would probably end up in a close enough position to disregard the stochasticity. On complex real world tasks, you might see a significant amount of stochasticity, which hurts the argument of this paper. The restriction is therefore quite strong and seems somewhat core to the proofs. I do not see a simple way to extend the work to include stochastic transition matrices. It seems that the metric changes according to how the Q functions were trained. If I understand correctly, the measurement is this: - A few Q functions are trained according to different hyperparameter sets - They are all evaluated on policy to obtain the "true average return" - They are also evaluated on some off-policy data to obtain the metric values - The correlation between the true average return and metric values are obtained. This implies the evaluation of novel techniques in this framework depend heavily on the set of Q functions trained. Thus, it's hard for new works to reproduce the results here, since if the choice of algorithm changes, then so might the ranking among algorithms. I'm confused to why some of the baselines have a strong negative correlation with the true average return. I wonder if the authors can analyze this.

[Author Response · NeurIPS 2019]

**All Reviewers:** Thank you for the review. A concern shared among reviewers was the focus on deterministic dynamics
and sparse reward. This simplification was only used for the theoretical analysis. **We can extend our result to the**
**stochastic case, and we present an empirical validation of our method on stochastic tasks.**

While we cannot include a full proof for the stochastic case, a proof sketch follows. We use the term *feasible* to denote
$(s, a)$ tuples from which $\pi^*$ has a non-zero chance of receiving positive reward. Let $c(s, a)$ be the probability that
stochastic dynamics bring a feasible $(s, a)$ to a catastrophic state, and let $c = \max_{s,a} c(s, a)$. This $c$ adds to existing
per-timestep error $\epsilon_t$. This gives $1 - R(\pi) \leq \sum_{t=1}^{T} (\epsilon_t + c) \prod_{i=1}^{t-1} (1 - (\epsilon_i + c)) \leq T(\epsilon + c)$, giving $R(\pi) \geq 1 - T(\epsilon + c)$.
When $\pi_b$ succeeds, we still get positive labels for each $(s, a)$ visited, so labels in the off-policy dataset are unchanged
and OPC can be estimated the same way. Stochastic dynamics only influence the lower bound on return.

Second, we empirically find that OPC performed well on a real-world robotic grasping task, which is necessarily
stochastic because the real world is stochastic. However, to further support this, **we ran new stochastic dynamics**
**experiments.** We modify the Tree environment to execute a random action instead of the policy's action with probability
$\epsilon$. We modify Pong to use sticky actions, a standard protocol for stochastic dynamics in Atari games introduced by
Machado et al., 2017. With small probability, the environment repeats the previous action instead of the policy's action.
Everything else is unchanged. In more stochastic environments, all metrics drop in performance since $Q(s, a)$ has less
control over return, but OPC and SoftOPC consistently correlate better than the baselines.

| | Stochastic Tree 1-Success Leaf | | | | | | | | Pong Sticky Actions | | | |
| | $\epsilon = 0.2$ | | $\epsilon = 0.4$ | | $\epsilon = 0.6$ | | $\epsilon = 0.8$ | | Sticky 10% | | Sticky 25% | |
| | $R^2$ | $\xi$ | $R^2$ | $\xi$ | $R^2$ | $\xi$ | $R^2$ | $\xi$ | $R^2$ | $\xi$ | $R^2$ | $\xi$ |
|---|---|---|---|---|---|---|---|---|---|---|---|---|
| **TD Err** | 0.01 | -0.11 | 0.01 | -0.07 | 0.00 | -0.05 | 0.00 | -0.05 | 0.05 | -0.16 | 0.07 | -0.15 |
| $\sum \gamma^t A^\pi$ | 0.00 | 0.06 | 0.00 | 0.01 | 0.01 | -0.07 | 0.00 | -0.02 | 0.04 | -0.29 | 0.01 | -0.22 |
| **MCC Err** | 0.09 | -0.31 | 0.07 | -0.27 | 0.01 | -0.06 | 0.01 | -0.11 | 0.02 | -0.32 | 0.00 | -0.18 |
| **OPC** | 0.18 | 0.46 | 0.13 | 0.38 | 0.01 | 0.08 | 0.03 | 0.19 | **0.48** | **0.73** | **0.33** | **0.66** |
| **SoftOPC** | **0.19** | **0.48** | **0.14** | **0.39** | **0.03** | **0.18** | **0.04** | **0.20** | 0.33 | 0.67 | 0.16 | 0.58 |

Third, regarding sparse rewards. **We can train with dense rewards at train time, as long as sparse binary rewards**
**are used at evaluation time.** This lets us support success vs failure tasks where reward shaping is added to speed up
learning. We can also extend our analysis to arbitrary rewards, by reducing all MDPs to non-negative sparse reward
MDPs, observing that $\mathbb{E}[X] = \int_0^\infty P(X \geq x) \, dx$ when $X$ is non-negative, and estimating each $P(X \geq x)$ with
PU-learning. We will attempt to complete this analysis for the final.

**R1:** We provided source code for OPC in the Tree env, but can add pseudocode to the paper as well.

**R2:** Thank you for the comments on novelty. We can work on clarifying how PU-learning connects to OPE, as this
is key to understanding our result. For the distribution mismatch assumption, in Appendix F we compared SoftOPC
performance in Sim Grasping and Real Grasping with different behavior policy datasets. In Real Grasping, correlation
was still strong when the behavior policy success rate was 28%, 40%, or 51%. In Sim Grasping, correlation was still
strong when the behavior policy success rate was 1% or 60%. We do find that OPC scores changed based on the
behavior policy, as expected, but as long as OPC scores are only compared using the same behavior policy dataset, the
relative rank of scores is mostly consistent across different datasets.

We also made sure to test how well OPC evaluates agents of widely varying performance, since those agents will have
very different state-visitation frequencies. In both Pong and Grasping, agents ranged from about 10% success to 90%
success. OPC was able to predict returns for all such agents with good correlation. As noted, we do not statistically
bound error from distributional mismatch like prior OPE work, but we do aim to empirically show robustness to
distribution mismatch. For data efficiency, we have found that about 100 episodes is sufficient to estimate OPC well.

**R3:** The $\gamma = 1$ restriction is an eval-time assumption to ensure total episode return is either 0 or 1. We use $\gamma < 1$ at
train time and $\gamma = 1$ for return at eval time.

Your description for how correlations are computed is correct. To minimize risk of overfitting results to specific learning
algorithms or Q-functions, we made sure to train many $Q(s, a)$ with many different learning algorithms. Q-functions
are sampled randomly (in Tree), trained with DQN or Double DQN (in Pong), with offline batch RL or on-policy RL
(in Pong and Grasping), or with different sim-to-real learning methods (in Real Grasping). In total, we used over 15
different Q-learning algorithms and hyperparameter settings. The OPC scores for models trained with each algorithm
were directly compared against each other. We found OPC to be predictive of return even in this setting, giving us
confidence that OPC is robust to the learning algorithm used. For reproduction, we list the exact algorithms used in
Appendix E, and include Tree environment source code in the supplement.

We believe baselines perform poorly because they measure how well $Q(s, a)$ fits the data, and the policy
$\pi(s) = \arg\max_a Q(s, a)$ can have high return even when $Q(s, a)$ does not fit the data well. We refer to Figure
1a and Section 5 for a more detailed explanation.

[Meta-Review · NeurIPS 2019]

The reviewers agreed that this is a nice paper that takes a highly novel approach to the important problem of off-policy evaluation. The main concern is how restrictive the problem setting is, but this is at least somewhat alleviated by the discussion and additional results in the author's rebuttal. The authors are strongly encouraged to integrate this into the final version. There were also minor concerns about readability and the lack of structure in the paper, so the authors are also strongly encouraged to significantly improve this in the final version.